# Spatial and Temporal Variabilities of PM$_{2.5}$ Concentrations in China Using Functional Data Analysis

**Deqing Wang [1], Zhangqi Zhong [2], Kaixu Bai [3] and Lingyun He [1,*]**

[1] School of Management, China University of Mining and Technology, Daxue Road 1, Xuzhou 221116, Jiangsu, China; dekinywang@cumt.edu.cn

[2] School of Economics, Zhejiang University of Finance & Economics, Hangzhou 310018, Zhejiang, China; zzhongz@zufe.edu.cn

[3] Key Laboratory of Geographic Information Science (Ministry of Education), East China Normal University, Shanghai 200241, China; kxbai@geo.ecnu.edu.cn

[*] Correspondence: Lingyun_he@cumt.edu.cn

**Abstract:** As air pollution characterized by fine particulate matter has become one of the most serious environmental issues in China, a critical understanding of the behavior of major pollutant is increasingly becoming very important for air pollution prevention and control. The main concern of this study is, within the framework of functional data analysis, to compare the fluctuation patterns of PM$_{2.5}$ concentration between provinces from 1998 to 2016 in China, both spatially and temporally. By converting these discrete PM$_{2.5}$ concentration values into a smoothing curve with a roughness penalty, the continuous process of PM$_{2.5}$ concentration for each province was presented. The variance decomposition via functional principal component analysis indicates that the highest mean and largest variability of PM$_{2.5}$ concentration occurred during the period from 2003 to 2012, during which national environmental protection policies were intensively issued. However, the beginning and end stages indicate equal variability, which was far less than that of the middle stage. Since the PM$_{2.5}$ concentration curves showed different fluctuation patterns in each province, the adaptive clustering analysis combined with functional analysis of variance were adopted to explore the categories of PM$_{2.5}$ concentration curves. The classification result shows that: (1) there existed eight patterns of PM$_{2.5}$ concentration among 34 provinces, and the difference among different patterns was significant whether from a static perspective or multiple dynamic perspectives; (2) air pollution in China presents a characteristic of high-emission "club" agglomeration. Comparative analysis of PM$_{2.5}$ profiles showed that the heavy pollution areas could rapidly adjust their emission levels according to the environmental protection policies, whereas low pollution areas characterized by the tourism industry would rationally support the opportunity of developing the economy at the expense of environment and resources. This study not only introduces an advanced technique to extract additional information implied in the functions of PM$_{2.5}$ concentration, but also provides empirical suggestions for government policies directed to reduce or eliminate the haze pollution fundamentally.

**Keywords:** PM$_{2.5}$ concentrations; functional principal component analysis; adaptive clustering analysis; functional ANOVA; spatial and temporal difference

## 1. Introduction

With the rapid development of industrialization and urbanization in China, haze pollution characterized by particulate matter smaller than 2.5 μm occurs more frequently and widely, which has seriously endangered the physical and mental health of residents, and threatened the sustainable

development of China's economy. According to statistics, the severe haze events that occurred in the first quarter of 2013 affected about 13.5% of the land area and 800 million people in China [1]. It is estimated that without a pollution control policy, the particulate matter pollution in China will lead to a 2% GDP loss and 25.2 billion USD in health expenditure in 2030 [2]. Thus, the prevention and control of haze pollution is not only a major livelihood project, but also an important way to assist the transformation of China's economic development model and the optimization and adjustment of China's economic structure. Since China's State Council released the "Air Pollution Prevention and Control Action Plan" in September 2013, which was a milestone for reducing $PM_{2.5}$ concentrations, local governments have promulgated their own air pollution control action plans. However, due to the multiple effect of various complex factors, such as an extensive development mode, unbalanced industry structure, and inefficient energy utilization, the fluctuations of $PM_{2.5}$ concentrations in different provinces exhibits obvious regional disparities and temporal characteristics [3–8]. Therefore, understanding the dynamic behavior of $PM_{2.5}$ concentrations is beneficial to further formulate and implement targeted environmental protection policies.

As a developing country with a dual structure, China is characterized by an unbalance of regional economic development and deteriorating environmental problems which resulted from its extensive mode of economic development and over-consumption of energy. Many researchers have pointed that haze pollution has become an obstacle for China to attract foreign investment, talent and tourists, and even threatens sustainable development in China [9,10]. Since $PM_{2.5}$ concentrations always change with time and fluctuate diversely across regions, intensive studies have been carried out on interpreting the spatial and temporal variability of $PM_{2.5}$ concentrations in China, both from city-level and national-scale perspectives. For example, taking Weifang city as a research object and based on the data of controlled monitoring stations, Li et al. concluded that the annual $PM_{2.5}$ concentrations reached a peak in 2013, while the seasonal and monthly $PM_{2.5}$ concentrations formed a U-shaped trend [11]. Considering Beijing and six surrounding cities as main research areas and based on correlation analysis of geo-statistics techniques, Zhai et al. studied the relevant relationship of $PM_{2.5}$ concentrations in Beijing [12] and found that the pollutant concentrations exhibit obvious cyclical fluctuation patterns with significant spatial correlation. Studies on spatial-temporal characteristics of $PM_{2.5}$ concentration on the national-scale includes references [13–16], their common conclusions are that China's haze pollution presented an obvious spatial spillover effect, and that $PM_{2.5}$ emissions had strong positive spatial autocorrelation with a certain spatial heterogeneity.

In light of the fact that $PM_{2.5}$ concentrations are the combined result of various factors, numerous literatures focus on exploring its primary cause via advanced methods. For example, Guan et al. presented an interdisciplinary study to measure the magnitudes of socio-economic factors in driving primary $PM_{2.5}$ emission changes in China between 1997–2010 [17]. According to the latest air quality standards of China, Wang et al. characterized the spatial and temporal variations of the concentrations of PM10, $PM_{2.5}$ and PM1 in China, their conclusion showed that the ratios of $PM_{2.5}$ to PM10 showed a clear increasing trend from northern to southern China, and that both emissions and meteorological variations dominate the long-term PM concentration trend, while meteorological factors played a leading role in the short term [18]. In order to monitor $PM_{2.5}$ by remote sensing in the Yangtze delta, Xu and Jiang constructed a $PM_{2.5}$ concentration model based on MODIS AOT, $PM_{2.5}$ concentration data of the 36 ground air quality observation sites and meteorological data, and empirical results proved their model estimation was higher than classical methodology [19]. Through the CAMx model, Cheng et al. examined spatial-temporal variations of $PM_{2.5}$ concentrations during two alerts based on multiple data sources, their results suggested that the implementation of emission reduction measures 1–2 days before red alerts could lower the peak of $PM_{2.5}$ concentrations significantly [20]. Using $PM_{2.5}$ concentrations data at China's provincial level over 1998–2012, Shao et al. adopted a dynamic spatial panel model and SGMM to empirically identify the key determinants of smog pollution, their results indicated that there was a significant U-shape curve relationship between smog pollution and economic growth, and smog pollution was worsening with economic growth in most eastern provinces [16]. With

PM10 and PM$_{2.5}$ concentration data collected from five air-quality monitoring sites in Lanzhou from October 2014 to October 2015, Guan et al. investigated the primary transport path using Hybrid Single Particle Lagrangian Integrated Trajectory Model (HYSPLIT) and the PM$_{2.5}$-to-PM10 ratio model [21]. Noticeably in these studies, all model constructions and empirical results were based on discrete and equal-sampled observations without any error disturbance. Additionally, the spatial-temporal characteristics of the PM$_{2.5}$ concentrations are also the major issues for air pollution investigations in many other countries, including developing and developed countries or regions. An array of literature focuses on assessing PM$_{2.5}$ spatial-temporal variability. For example, based on data from biophysical remote sensing and GIS, Famoso F, et al. conducted the measurement and modeling of ground-level ozone concentration of Catania in Italy [22]. Using PM$_{2.5}$ concentrations at 71 EPA monitoring stations from 2006 to 2011, Wu et al. applied a hybrid kriging/LUR model to assess the spatial-temporal variability of PM$_{2.5}$ for Taiwan [23]. In order to identify the local and long-range sources of PM$_{2.5}$ and their relationships with other air pollutants and meteorology, Mukherjee et al. investigated the local and distant sources of PM$_{2.5}$ from 2014 to 2017 in Varanasi city located in middle Indo-Gangetic plain (IGP) of India using various statistical modeling methods [24], such as conditional bivariate probability function (CBPF), land use regression (LUR) and trajectory statistical models (TSM) like potential source contribution function (PSCF),concentration weighted trajectory (CWT) and trajectory cluster analysis. Considering LUR models may fail to capture complex interactions and non-linear relationships between pollutant concentrations and land use variables, Brokamp et al. developed a novel land use random forest (LURF) model and compared its accuracy and precision to a LUR model for elemental components of PM in the urban city of Cincinnati, Ohio [25]. The comprehensive comparison showed that these methodological approaches provide efficient means to better assess PM$_{2.5}$ spatial-temporal variations and prediction levels, and usually work well with large scale pollution dispersion.

Although the existing studies on PM$_{2.5}$ concentrations have provided many meaningful suggestions, their shortcomings are also obvious. Firstly, most of the empirical methods were statistical descriptions or econometric modeling using discrete noisy data, which cannot mine the continuous trajectory and dynamical information implied in the changing process of PM$_{2.5}$ concentrations. Secondly, most studies focused on the research scale of mainland China and metropolitan areas which neglected the increase in regional differentiation, or analyzed the individual district separately, with little consideration of the homogeneity of different regions. Thirdly, the existing studies used mostly rough and historical data collected by ground monitoring stations. Unlike the air pollution index, PM$_{2.5}$ concentrations have only been recorded since 2012 in China, thus having too short or too old time scales that result in a low temporal resolution.

It should be noted that data in many scientific experiments are recorded repeatedly through time or space and have been seen to arise as a continuous process. Examples of such kinds of observations are hourly records of PM$_{2.5}$ concentrations and daily records of air quality. The classical discrete data modeling approaches are found to be inadequate in understanding the underlying process of the pollutant and hence prevent the implicit information from being revealed [26,27]. The coming era of big data makes it possible to analyze these discrete noisy data by converting them into continuous and smoothing functions, then we can explore the dynamic information implied in the original data from multiple derivative functions [28]. The new modern statistical methodology which considers discrete time point values as observations of continuous functions over a continuum is termed as *Functional Data Analysis* (FDA) [29]. The functional concept may bring additional insight by looking at the pattern and temporal variation of pollutant variables in the form of smoothing curves or functions. A previous study by Shaadan et al. highlighted the advantages of an FDA approach in assessing and comparing the PM10 behavior [27], while several studies that focus on using FDA to analyze the pollution behavior have proved the merits of FDA in environmental pollution research [27,30–32]. To the best of our knowledge, there is little research studying the spatial-temporal variability of PM$_{2.5}$ concentrations in China within the framework of continuous functions. Thus, using PM$_{2.5}$ concentrations data at

provincial level from 1998 to 2016, this study will employ FDA to classify the fluctuation patterns of PM$_{2.5}$ pollution for 34 provinces, and dynamically compare their evolving trajectories. The empirical results is helpful for enhancing the recognition of the spatial distributions and dynamic changes of PM$_{2.5}$ concentrations in China, and can provide quantitative support for governments to formulate and implement air pollution prevention and control measures.

## 2. Methodology

In this subsection, we introduce the framework of FDA, which mainly includes smoothing PM$_{2.5}$ pollution functions with roughness penalty, classifying categories of fluctuations via adaptive weighting clustering analysis, and testing the significance of difference among different regions using functional ANOVA. Data processing and analysis are conducted using the free R software (R Development Core Team, 2018), together with package "*fda.usc*" (Febrero-bande et al., 2016) [33] and package "*fda*" (Ramsay et al., 2013) [34].

### 2.1. Smoothing with or without Roughness Penalty

PM$_{2.5}$ concentrations data is often recorded at discrete time intervals, and is usually analyzed within the framework of traditional time series or multivariate statistical approaches. But in the context of functional data analysis, the PM$_{2.5}$ concentration data is essentially assumed to be continuous with time, even though the concentration data is collected at a daily, monthly or annual frequency. The primary goal of FDA is to convert discrete data, such as $y_{i1}, \cdots, y_{iT_i}$, to a smooth function $f_i(t_j)$, which is computable for any values of $t_j$ with $j = 1, \cdots, T_i$. There are two ways to convert the discrete data into continuous functions, their core difference lies in the presence or absence of disturbance factors. If the data is assumed to be errorless, that is $y_{ij} = f_i(t_j)$, the interpolation method may be employed. However, if there are observational errors that need removing, the smoothing process will be used. In reality, the PM$_{2.5}$ concentrations data is always contaminated by random noise $\varepsilon_{ij}$, that is $y_{ij} = f_i(t_j) + \varepsilon_{ij}$. Considering the universality of practical problems and our intention of converting the discrete noisy data into quadratic differentiable functions, we mainly discuss the smoothing functional method with roughness penalties to error disturbances. Assuming $\mathbf{\Phi}(t) = \{\phi_1(t), \cdots, \phi_L(t)\}$ to be the optimal basis function in Hilbert space, the *sum of squared fitting residuals for the roughness penalty* ($PENSSE_\kappa$) [29,34,35] is given as follows:

$$PENSSE_\kappa = \sum_{i=1}^{n} \{\sum_{j=1}^{T_i} [y_{ij} - f_i(t_j)]^2 + \kappa \int_T [f_i''(t)]^2 dt\} \qquad (1)$$

The intrinsic continuous function $f_i(t)$ in Equation (1) is a linear approximation of the basis function to meet the criterion of minimizing the $PENSSE_\kappa$, i.e., $f_i(t) = \sum_{l=1}^{L} \beta_{il}\phi_l(t)$, where $\beta_{il}$ denotes the coefficients of the basis function expansion. The smoothing parameter $\kappa$ specifies the proportion between the goodness of model fitting and the smoothing amount of the function curve. Large values of $\kappa$ will increase the amount of smoothing. The best value for the smoothing parameter $\kappa$ is determined by the minimum generalized cross-validation $GCV(\kappa)$ [34]. The criterion is given as follows:

$$GCV(\kappa) = \left(\frac{n}{n - df(\kappa)}\right)\left(\frac{PENSSE_\kappa}{n - df(\kappa)}\right) \qquad (2)$$

where the degree of freedom $df(\kappa) = trace\left\{\mathbf{\Phi}(\mathbf{\Phi}'\mathbf{\Phi} + \kappa\mathbf{R})^{-1}\mathbf{\Phi}'\right\}$ and the roughness penalty matrix $\mathbf{R}$ is expressed as $\mathbf{R} = \int D^2\phi(s) \cdot D^2\phi'(s)ds$. Based on the above symbols, solving Equation (1) for $\mathbf{\beta}$ will give us $\hat{\mathbf{\beta}} = (\mathbf{\Phi}'\mathbf{\Phi} + \kappa\mathbf{R})^{-1}\mathbf{\Phi}'\mathbf{y}$, then 34 provinces with 18 yearly measurements will be transformed into 34 PM$_{2.5}$ concentrations curves. A complete theoretical review of the penalty smoothing method can be found in Kokoszka et al. (2017) [36], and the steps of the algorithm are detailed in Ramsey et al. (2009) [34]. It should be noted that FDA does not restrict all samples to be sampled at regular intervals or

same frequency on the observing interval, that is $T_i \neq T_j$. Thus, the relaxed structure of data collection and hypothesis of distribution enable FDA to depict practical problems more comprehensively and flexibly [37]. Particularly, once the intrinsic functions are reconstructed from the discrete noisy data, we can not only display the continuously changing trajectory of PM$_{2.5}$ concentrations statically from the holistic perspective, but also can analyze their dynamic process interactively from multiple derivative functions.

### 2.2. Significance Test of Difference via Functional Analysis of Variance

The functional analysis of variance (F-ANOVA) is used to test whether two or more sets of functional data are identical, independent, and come from the same population. The verification was done by comparing their functional means. Let $g$ represent the number of groups or zones, with $f_{ij}(i = 1, \cdots, g; j = 1, \cdots, n_i)$ as the $j$th-functional data for $i$ groups, and $n_i$ is the number of curves in group $i$. As a first step in F-ANOVA, the classical F statistic in the form of functional data is considered and is given as:

$$F_n = \frac{\sum_{i=1}^{g} n_i \left\| \overline{f_{i.}} - \overline{f_{..}} \right\|^2 / (g-1)}{\sum_{i,j} \left\| f_{ij} - \overline{f_{i.}} \right\|^2 / (n-g)} \tag{3}$$

where $\|\cdot\|$ denotes the usual $L^2$ norm as $\|f\| = (\int f^2(t)dt)^{1/2}$. The expressions used in Equation (3) are described by $f_{ij} = (f_{ij}(t_1), \cdots, f_{ij}(t_T))\prime$, $\overline{f_{i.}} = (\overline{f_{i.}}(t_1), \cdots, \overline{f_{i.}}(t_T))\prime$ and $\overline{f_{..}} = (\overline{f_{..}}(t_1), \cdots, \overline{f_{..}}(t_T))\prime$, which can be computed as $\overline{f_{i.}}(t) = \sum_{j}^{n_i} f_{ij}(t)/n_i$, $n = \sum_{i=1}^{g} n_i$ and $\overline{f_{..}}(t) = \sum_{i=1}^{g} n_i \overline{f_{i.}}(t)/n$. $\overline{f_{..}}$ is the global functional mean and $\overline{f_{i.}}$ is the functional mean in the $i$th groups, respectively, at time $t$. With the above symbols, the equivalent statistic of Equation (3) can be rewritten as:

$$V_n = \sum_{i<j}^{g} n_i \left\| \overline{f_{i.}} - \overline{f_{j.}} \right\|^2 \tag{4}$$

Given the null hypothesis of having the same functional means for each $i$ group, that is, $H_0 : \overline{f_{1.}} = \cdots = \overline{f_{g.}}$, calculate the critical value $P_{H_0}\{F > F_{n,\alpha}\} = \alpha$ and $P_{H_0}\{V > V_{n,\alpha}\} = \alpha$ at the specified significance level $\alpha$ respectively. $H_0$ should be rejected if the variability between groups, which are measured by the difference in the sample means $F_n$ and $V_n$, is large enough to be expressed as $F_n > F_{n,\alpha}$ and $V_n > V_{n,\alpha}$. In other words, the test is found to be statistically significant if the $p$-value is less than the $\alpha$ significance level. The detailed steps of algorithm can be found in Cuevas et al. (2004) [38]. This procedure uses a point-wise critical value obtained using a permutation test for reference lines [39].

### 2.3. Functional Principal Component and Adaptive Clustering Analysis

The intrinsically infinite dimensionality of functional data poses challenges to traditional clustering methods used for classifying discrete data, both for theory and computation [40–42]. In order to reduce the cost of calculation and elevate the accuracy of classification, we employ the adaptive weighting clustering analysis to classify the fluctuation patterns of PM$_{2.5}$ concentrations curves, and use bootstrap sampling methods to test the significance and robustness of difference among groups.

Let $V(s,t) = (N-1)^{-1}\sum_{i=1}^{N} [f_i(s) - \overline{f}(s)][f_i(t) - \overline{f}(t)]$ be a continuous covariance operator on $[0, T]^2$, by Mercer's lemma [43], there exists an series of orthogonal functions $\varphi_k(t)$ with their corresponding non-negative decreasing eigenvalues $\lambda_k$ satisfying:

$$\int_0^T V(s,t)\varphi_l(s)ds = \lambda_l \varphi_l(t) \qquad t \in [0, T], l \in N \tag{5}$$

with respect to

$$\int_0^T \varphi_l(t)\varphi_m(t)dt = \delta_{lm} = \begin{cases} 1, & m = l \\ 0, & m \neq l \end{cases} \tag{6}$$

Further, for the second-order continuous stochastic process $\{f(\cdot), t \in [0,T]\}$ on $L^2(T)$, the realization of the process for the $i$th subject is $f_i(t)$. Denote $\mu(t)$ and $V(s,t)$ as the mean and covariance of $f_i(t)$, respectively. Then the Karhunen-Loève expansion of $f_i(t)$ [44] is given as:

$$f_i(t) = \mu(t) + \sum_{k=1}^{\infty} \zeta_{ik}(f_i)\varphi_k(t) \;, \quad t \in [0,T] \tag{7}$$

where $\zeta_{ik}(f) = \int_T (f_i(t) - \mu(t))\varphi_k(t)dt$ are the functional principal components (FPCs), sometimes referred to as *scores*. The $\zeta_{ik}(\cdot)$ are independent across $i$ for a sample of independent trajectories and are uncorrelated across $k$ with $E(\zeta_{ik}) = 0$ and $\text{var}(\zeta_{ik}) = \lambda_k$. Furthermore, the covariance of $\zeta_{ik}(\cdot)$ satisfies

$$E[\zeta_k(f)\zeta_l(f)] = \lambda_k \delta_{kl} \quad k, l \in N \tag{8}$$

From the Karhunen-Loève expansions of stochastic process, we can infer that $\zeta_{ik}(f)$ are the projection scores of centered functions $(f_i(t) - \mu(t))$ to the direction of a standard orthogonal basis function $\varphi_k(t)$, which is objectively derived from the information implied in original PM$_{2.5}$ concentrations data. Based on the Karhunen-Loève expansion of Equation (7), the difference among categories of different functional data is entirely reflected by the difference between their projected scores $\zeta_{\cdot k}(f)$. Since $\lambda_k$ is also the variance of $\zeta_{\cdot k}(f)$, and without loss of generality, assume their sequence order satisfying $\lambda_1 \geq \lambda_2 \geq \cdots \geq 0$. In order to reflect the objective difference of classification information implied in $\zeta_{\cdot k}(f)$, define $\beta_k = \lambda_k / \sum_{l \geq 1} \lambda_l$ as the weight of $\zeta_{\cdot k}(f)$, we reconstruct the adaptive weighting distance between $\zeta_i(f)$ and $\zeta_j(f)$ as:

$$d[f_i(t), f_j(t)|q] = [\sum_{l=1}^{\infty} (\beta_l |\zeta_l(f_i) - \zeta_l(f_j)|)^q]^{\frac{1}{q}} \tag{9}$$

The distance parameter $q$ is analogous to the classical definition of similarity, with $q = 2$ corresponding to the Euclidean distance. In practice of conducting adaptive clustering analysis, it is unnecessary to choose all the FPCs. Without a loss of core information, the criteria for selecting the number of FPCs is the minimum value $M$ that reaches a certain level of the proportion of total variance explained by the $M$ leading components, such as $\sum_{l=1}^{M} \lambda_l / \sum_{l \geq 1} \lambda_l 1_{\{\lambda_l > 0\}} \geq 90\%$. Further information on the theoretical foundation and applications of functional adaptive clustering method could be obtained from our previous works [45–47].

## 3. Data Sources and Empirical Results

### 3.1. Data Sources

The reliable data source of PM$_{2.5}$ concentrations is crucial for this study. After China's Ministry of Environmental Protection issued the new environmental air quality standard in February 2012, local governments began to routinely record and release the data of PM$_{2.5}$ concentrations. Due to lacking data of a long-term time span, it is difficult to extract the dominant patterns of evolution for PM$_{2.5}$ concentrations. Besides, because the number of ground monitoring stations is small and its distribution is uneven, the rough reflection using sparse points to denote the whole area cannot exactly measure the real situation of PM$_{2.5}$ concentrations. In order to solve the data deficiency of historical and regional PM$_{2.5}$ concentrations, this paper adopts the data sets regarding the raster data of the annual average PM$_{2.5}$ concentrations at a global level using satellite-based environmental surveillance, which is published by the socio-economic research center at Columbia University. The data sets used here are obtained from the study by van Donkelaar et al. (2016) [48], which had calibrated each AOD source using AERONET observations. Based on the data sets, using geographic information system technology, we could obtain the corresponding raster data of the annual average PM$_{2.5}$ concentrations in China for the period 1998–2016. Notably, however, compared with that directly from actual

monitoring data on the ground, although the data sets collected from satellite-based monitoring process could be affected by meteorological factors, which thereby led to a lower accuracy, the data sets from actual monitoring data on the ground could only roughly provide PM$_{2.5}$ concentrations in a region using area object other than point one based on point source data, and thus it's difficult to accurately measure global PM$_{2.5}$ concentrations in the region. Being an important non-point source data, satellite-based monitoring data sets have more advantages than the traditional methods in terms of reflecting the value of the PM$_{2.5}$ concentration and its change trend in a region. Actually, the research based on satellite-based monitoring data has won the recognition of the academics, owing to the works of Nordhaus et al. [49,50], who won the Nobel Prize Economics in 2018. Thus, the satellite-based monitoring data employed by this study is reliable. Additionally, from the technical perspective of empirical analysis, FDA owns the congenital advantage of modeling noisy data when smoothing with roughness penalty, even when the data is sparse or sampled unequally. Thus, having combined the reliable data source with the advanced methodology, it is reasonable to draw reliable conclusions.

*3.2. Reconstructing PM$_{2.5}$ Concentrations Functions and Summary Statistics*

As a rule of thumb, it is safer to smooth only when necessary if we want to retain the maximal information [51,52]. In order to verify the necessity of roughness penalty in reconstructing PM$_{2.5}$ concentrations functions, we firstly select the optimal smoothing parameter which minimizes the GCV. Figure 1 shows how the GCV criterion varies as a function of $\log_{10}(k)$ for the mean of PM$_{2.5}$ concentrations. The minimizing value of $k$ is found to be 1.25, and at that value $df(k) = 3.81 \approx 4$. Next, we plot the penalized PM$_{2.5}$ concentrations curve with the selected smoothing parameter, and the comparison object, that is the mean of un-penalized PM$_{2.5}$ concentrations curve without roughness penalty, is also plotted in Figure 2. Taking the trajectory of the penalized curve as benchmark, we can clearly see that the mean of PM$_{2.5}$ concentrations experienced a fluctuation, increased rapidly and then declined slowly, and reached its maximal value round 2007. Though there is a slight rebound during the descending process, the PM$_{2.5}$ concentrations kept a downward trend at the end of the research interval, which can be attributed to the synthetic effect of environmental protection policies [53]. In contrast, the trajectory of the un-penalized PM$_{2.5}$ concentrations curve fluctuated frequently with a cycle about every two years, but the dominant changing trend of PM$_{2.5}$ concentrations was obscured by those slight fluctuations with various amplitudes. Thus, we decided to smooth the PM$_{2.5}$ concentrations with a roughness penalty at the value of $k = 1.25$.

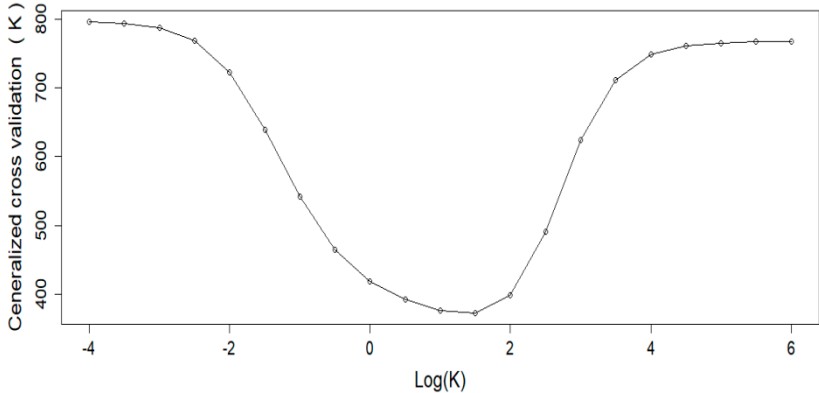

**Figure 1.** The values of generalized cross validation or GCV criterion for choosing the smoothing parameter for fitting the mean of PM$_{2.5}$ concentrations.

Figure 3 displays the summary statistics for the functional information of PM$_{2.5}$ concentrations in terms of their mean and standard deviation for all regions. It shows that generally, the highest mean PM$_{2.5}$ concentrations were recorded around 2007, the year during which environmental protection policies were formulated and implemented intensively in China, such as a campaign for energy-saving.

The trajectory of the standard deviation function also follows the same pattern as the functional mean of $PM_{2.5}$ concentrations. That is, the $PM_{2.5}$ concentrations variability increased rapidly since 1998, and reached its maximal value around 2007, then kept a high level with a slight rebound. It should be noted that the value of standard deviation is larger when the level of $PM_{2.5}$ concentrations is high. For the increasing deviation, we ascribe it to the differentiated reactions from different regions when facing the dilemma between environmental protection and extensive economic development.

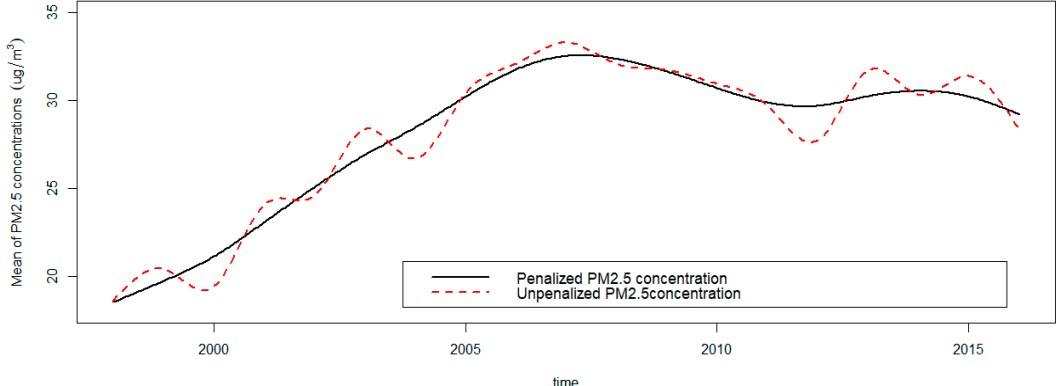

**Figure 2.** Smoothing curves with roughness penalty and without roughness penalty for the mean of $PM_{2.5}$ concentrations.

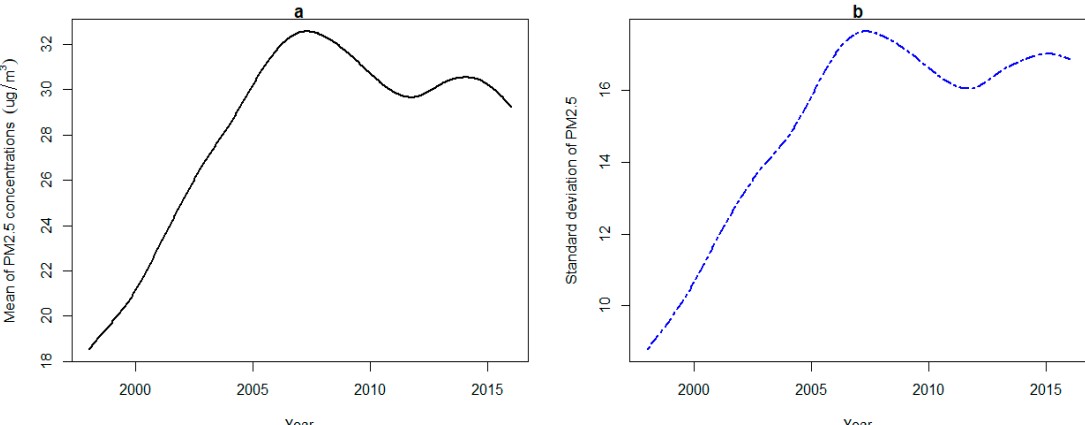

**Figure 3.** (**a**) The average of mean and (**b**) the standard deviations for yearly $PM_{2.5}$ concentrations curves of all provinces.

Information about the first and second derivatives from the smoothing function can give information on the rate of change and the acceleration in $PM_{2.5}$ concentrations according to time compared to the traditional multivariate statistical approaches which could not possibly capture this kind of information [24,25]. In order to dynamically analyze the evolving process of $PM_{2.5}$ concentrations from 1998 to 2016, we can extract more information by studying how derivatives relate to each other, which is often called a *phase-plane plot* (PPP) [54]. The energy transferring between the first order derivative of $PM_{2.5}$ concentrations which is called average velocity and the second order derivative which is called average acceleration, was shown in Figure 4. The numbers along the curve indicate the year of $PM_{2.5}$ concentrations. The trajectory of PPP exhibits several interesting features. There were two obvious cycles of energy transferring between velocity and acceleration, with the year 2007 as a landmark. During the first cycle, although the sign of growth acceleration for $PM_{2.5}$ concentrations alternated from positive to negative frequently, the growth velocity remained positive all the time, and the largest growth velocity occurred between 2001 and 2002. During the second cycle from 2007 to 2016, both the sign of growth velocity and acceleration alternated between positive and negative, with a larger oscillation. The first cycle corresponded to the period during

which the decoupling indicators of China's resources consumption and GDP growth is much lower. The key reason for this phenomenon is that China was in the process of industrialization, particularly in the process of heavy industrialization, which caused the rapid growth of infrastructure construction and consumed vast amounts of basic materials. The second cycle corresponded to the period during which the $PM_{2.5}$ concentrations fluctuated with a high frequency, due to the intensive formulation and implementation of environmental protection policies.

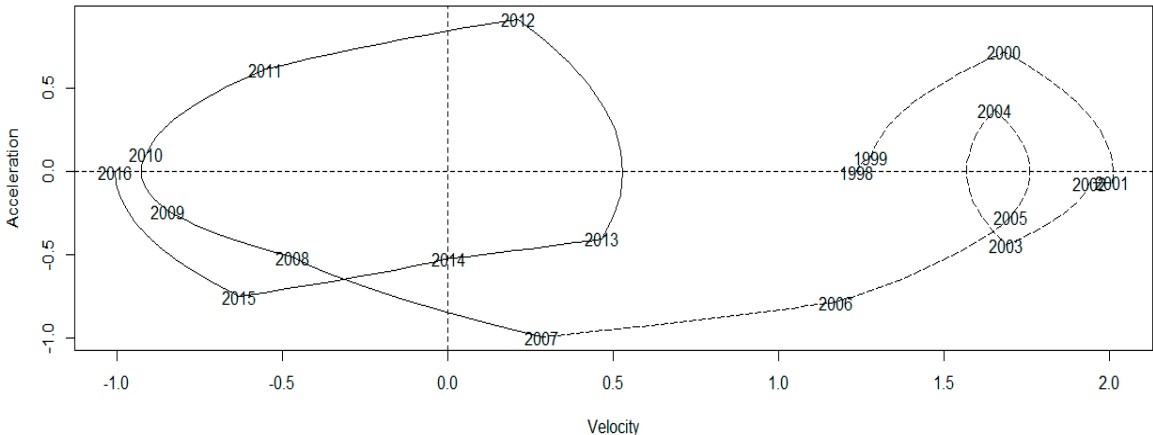

**Figure 4.** The phase-plane plot for the average $PM_{2.5}$ concentrations curve: the second derivative (acceleration) versus the first derivative (velocity).

### 3.3. Temporal Variability Decomposition

As one of the most important advantages for FDA, the temporal variance-covariance surface as well as the corresponding contour in functional data gives new ways to gather information, more than a single value or matrix obtained in the traditional univariate and multivariate contexts [55]. The estimated variance surface of $PM_{2.5}$ concentrations from 1998 to 2016 with its corresponding contour plot are presented in Figure 5. We can see the variability becoming larger and larger since 1998, and the highest variability occurs around 2007, the period which also corresponds to the highest mean of $PM_{2.5}$ concentrations. In order to further explore the potential variation from curve to curve, we employ functional principal components analysis (FPCA) to decompose the covariance function. Figure 6 displays the result of covariance decomposition via FPCA for $PM_{2.5}$ concentrations after varimax rotation. For each of the first three principal components, three curves are plotted. The solid curve is the overall smoothed mean which is the same in all provinces just for reference purposes, and the other two curves show the effect of adding and subtracting a suitable multiple of the principal component weight function. The accumulative percentage of variance explained by the first three components is 99.7%, indicating that there was almost no valuable information lost.

It can be seen that, each of the three principal component functions quantifies variability corresponding to a particulate period, thus the trajectories of the varimax rotated FPCs give good interpretations. Specifically, the first principal component function, which accounts for 69.2% of the total variation in the original $PM_{2.5}$ concentrations observations, mainly depicts the variability from 2003 to 2012. Actually, the period from 2003 to 2012 was called the "golden ten years" for the coal industry, which also are the "golden ten years" of China's rapid economic growth. However, restricted to various factors such as industrial structure and resources endowment, each province can only choose the suitable development mode according to its own situation. As a result, the emissions level of particulate matter for each province deviated greatly from the overall mean. Consequently, the covariance function of $PM_{2.5}$ concentrations among 34 provinces oscillated drastically during the period of fossil fuel energy being highly consumed. In contrast, the second and the third principal component function mainly reflect variability located at the end and beginning of the research period, respectively. The proportions of total variation they accounted for is nearly equal, that is 15% and

15.5%, which is even less than the one fourth of the amount explained by the first principal component function. In light of this, the vast disparity in variance contribution rate for each principal component function requires differentiated treatment when conducting functional clustering analysis on the scores of principal components.

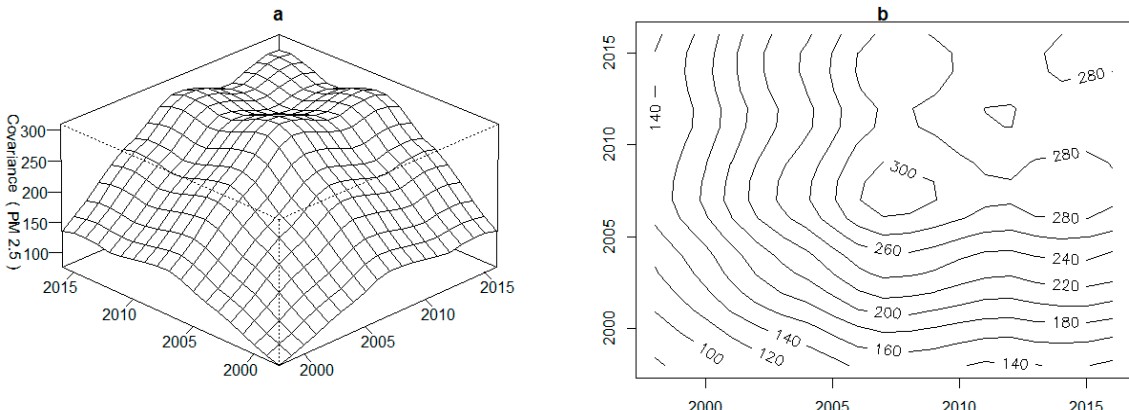

**Figure 5.** (**a**) Estimated variance surface of PM$_{2.5}$ from 1998 to 2016 and (**b**) the corresponding contour map.

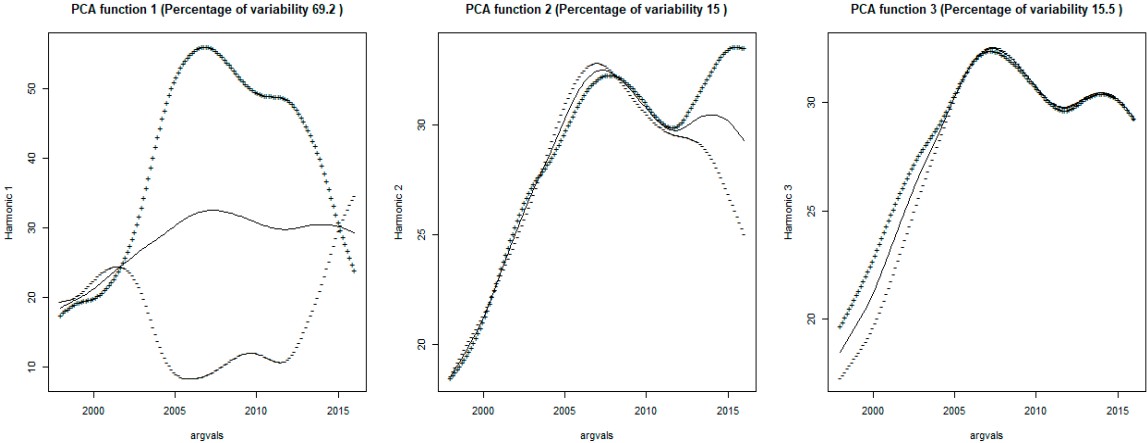

**Figure 6.** The first three varimax-rotated principal components of PM$_{2.5}$ concentrations.

### 3.4. Region Classification and Significance Test

In order to visually explore how curves clustering within the three-dimensional subspace spanned by the first three principal component functions, Figure 7 displays the scatter plots of scores on pairs of weight functions for each province. It shows that there is essentially no correlation among these scores, so the three principal components can be considered as uncorrelated variables within 34 provinces. Although the three scatter plots show no very distinctive features, the distribution range for each of the three component differs vastly. It can be seen that the scores on the first principal component ranges from about −100 to 150, with a considerable lager amount of variability. However, the scores on the other two components distribute with a nearly equal range, which is far less than that of the first component. In view of the vast disparity of information amount, different weights for the three principal components should be taken into account when employing clustering analysis to classify the categories of fluctuation.

As a preliminary step of unsupervised classification, it is necessary to determine the number of clusters before conducting adaptive weighting clustering analysis. The optimal number of clusters in unsupervised classification is still an open question [56]. In this study, we adopt the *wssplot*( ) and *NbClust*( ) functions to objectively choose the number of clusters [57]. The selecting criterion presented

in Figure 8 indicates that there is a distinct drop in the within-groups sum of squares when moving from one to eight clusters. After eight clusters, this decrease drops off, suggesting that an eight-cluster solution may be a good fit to the $PM_{2.5}$ concentrations data in 34 provinces. Besides, 14 of 24 criteria provided by the *NbClust* package suggest an eight-cluster solution. So we chose eight as the optimal number of clusters, and the initial classification via adaptive weighting clustering was listed in the second column of Table 1, the spatial distribution of $PM_{2.5}$concentrations for each group was illustrated in Figure 9.

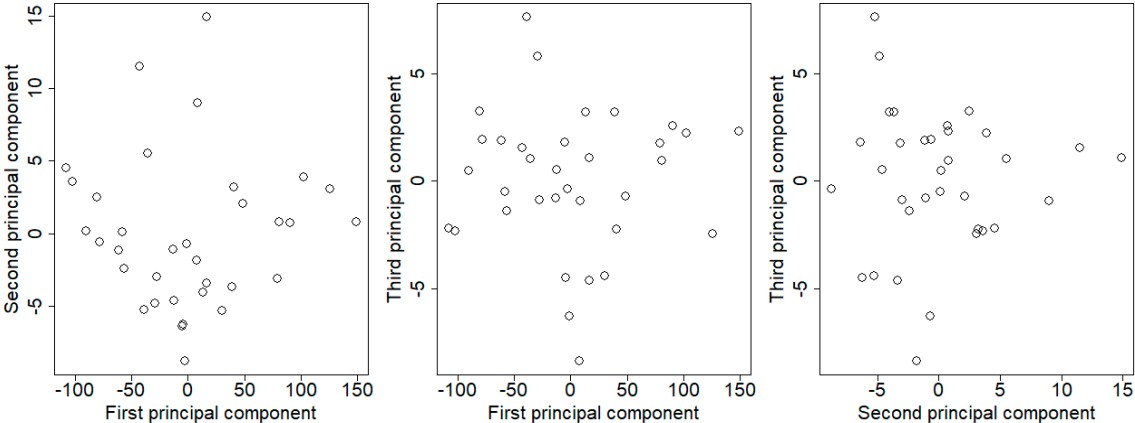

**Figure 7.** Plot of the first three principal components scores of $PM_{2.5}$ concentrations.

Although we have objectively classified the $PM_{2.5}$ concentrations curves of 34 provinces into eight clusters, it is necessary to quantitatively conduct a further test in the robustness of the initial classification. In other words, we should prove the hypothesis that there indeed was significant difference between the eight groups. To address the above problem, the F-ANOVA based on 1000 bootstrap sampling is performed on original functions as well as their velocity and acceleration, respectively. Figure 10 illustrated the test results of the original $PM_{2.5}$ concentration functions, and the robust test results corresponding to the first order and the second order derivatives were presented in Figures 11 and 12, respectively. Using the test results of F-ANOVA from Figures 10–12, we can safely draw the conclusion that, the fluctuation patterns between the eight groups of $PM_{2.5}$ concentration functions was significantly different at the level of 1%, whether from the static perspective or from multiple dynamic perspectives. Thus, on a credible quantitative analysis basis, we are confident in excavating more reliable and deeper information by further comparing the different trajectories of $PM_{2.5}$ concentration curves in each groups.

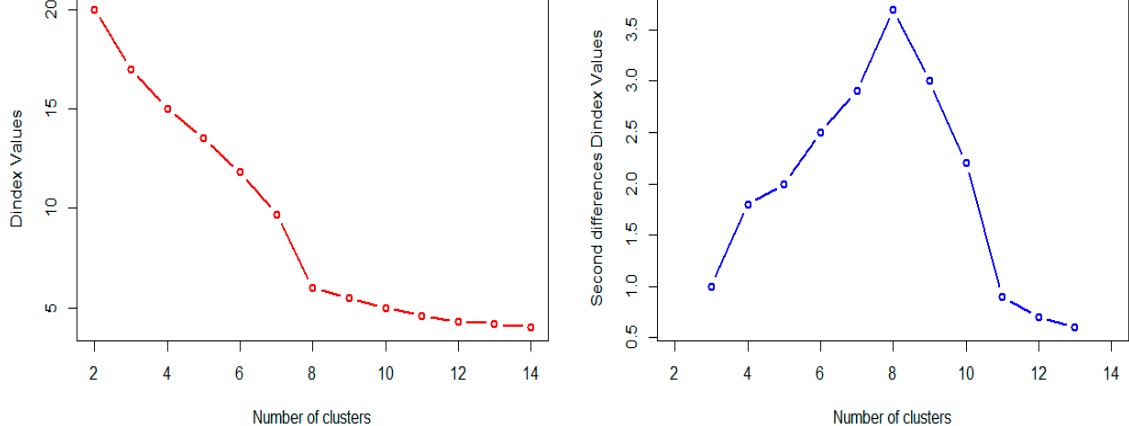

**Figure 8.** Dindex graphic for determining the best number of clusters.

**Table 1.** The classification of PM$_{2.5}$ concentrations fluctuation.

| Group | Provinces | Characteristics | Reasons |
|---|---|---|---|
| 1 | Liaoning, Jilin, Zhejiang, Guangdong, Guangxi | the dominant fluctuation pattern of PM$_{2.5}$ concentrations in China with slightly more than the national average level and a moderate deviation in the end stage | sparsely-populated provinces with developed heavy industry, or intensively-populated provinces of highly developed tertiary industry Pearl River Delta |
| 2 | Heilongjiang, Hainan, Sichuan, Yunnan | the second lowest level with a slightly growing trend and an increasing deviation | provinces with tourism as their pillar industry |
| 3 | Shanxi, Jiangxi, Chongqing, Guizhou, Hong Kong, Macao | the dominant fluctuation pattern of PM$_{2.5}$ concentrations in China with slightly less than the national average level | intensively-populated provinces with steady and humid atmospheric |
| 4 | Fujian, SHANXI, Ningxia, Taiwan | the third lowest level with a nearly constant deviation | provinces in the southeast coast of China strongly influenced by maritime monsoon, or provinces in western with stable atmospheric circulation throughout the year |
| 5 | Neimenggu, Tibet, Gansu, Qinghai, Xinjiang | the lowest level, without obvious growth or deviation. | sparsely-populated provinces in western frontier of China, with traditional agriculture and livestock farming |
| 6 | Tianjin, Shandong | the highest level and largest fluctuation amplitude, with obvious turning points corresponding to government environmental policies | energy-intensive industries with enriched, high-frequency use of diesel freight vehicles and non-road machinery |
| 7 | Shanghai, Jiangsu, Anhui, Henan | the second highest level, mainly located at Yangtze River Delta with obvious secondary pollution | the most active economic area in China, labor-intensive and enriched industries, resulting in a large quantity of fumes discharged from vehicles |
| 8 | Beijing, Hebei, Hunan, Hubei | the third highest level with a growing deviation | highly intensive-populated region, or inland region with secondary pollution from their surrounding neighborhood |

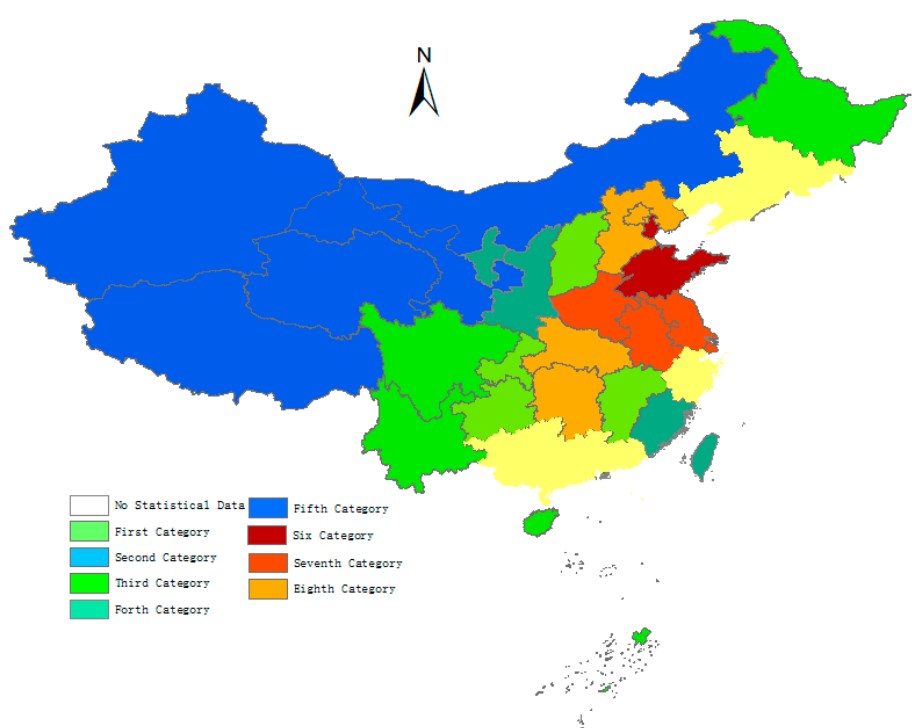

**Figure 9.** Spatial distribution of PM$_{2.5}$ concentrations for eight groups in China.

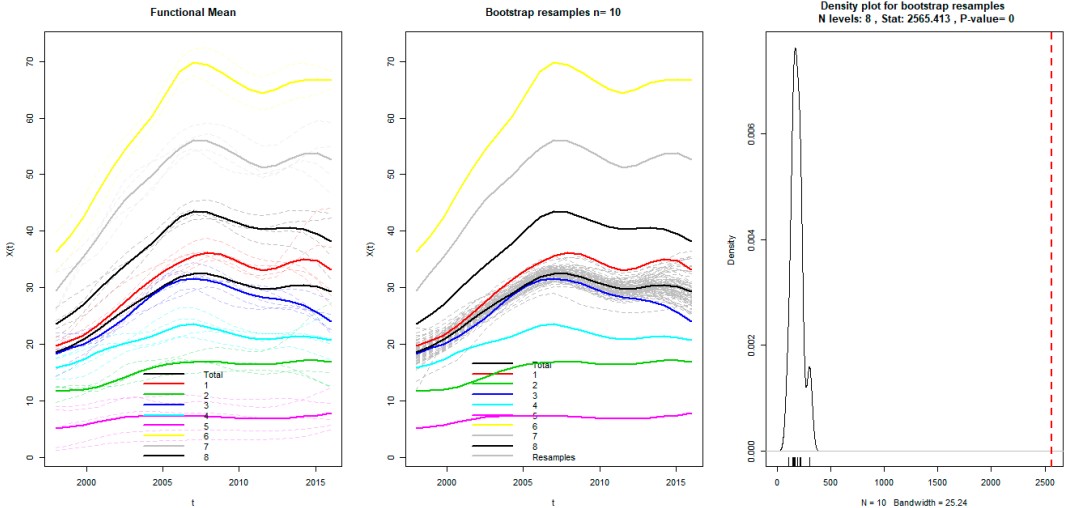

**Figure 10.** F-ANOVA test for absolute level of PM$_{2.5}$ concentration functions.

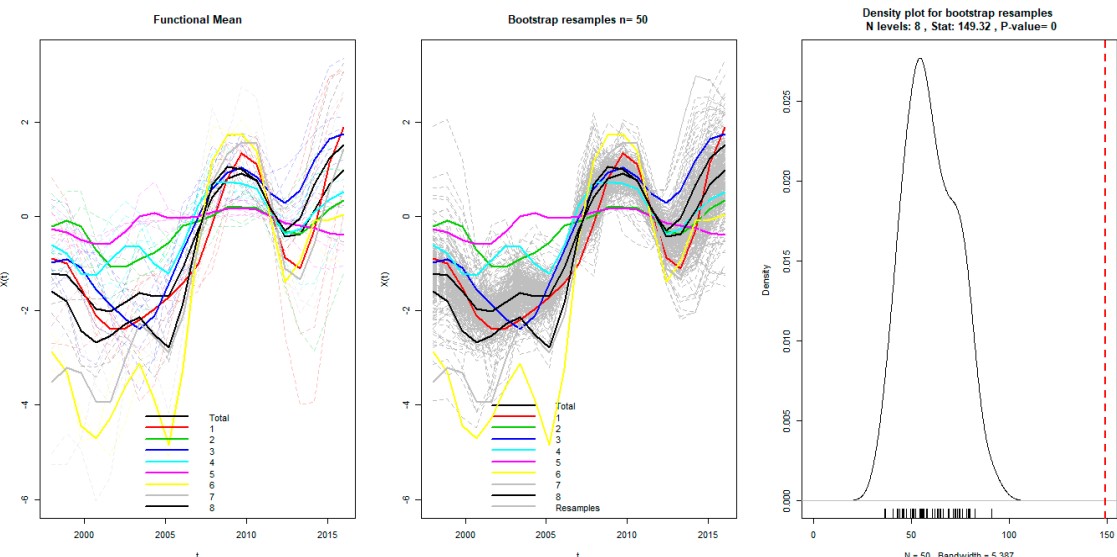

**Figure 11.** F-ANOVA test for the velocity of PM$_{2.5}$ concentration functions.

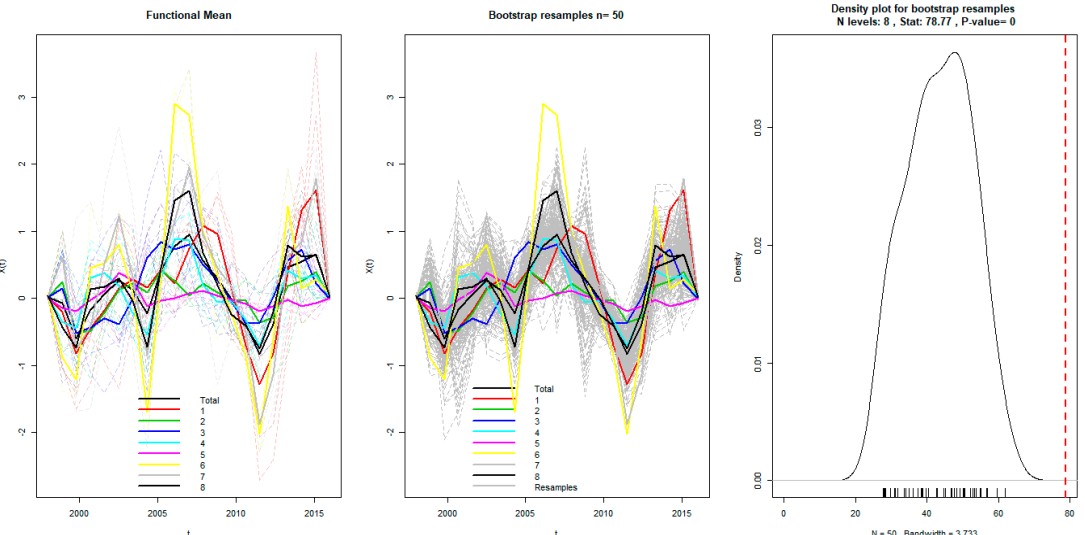

**Figure 12.** F-ANOVA test for the acceleration of PM$_{2.5}$ concentration functions.

### 3.5. Comparing the Fluctuation Patterns of PM₂.₅ Concentration in Each Group

Due to multiple differences in industrial structure and topography, together with the different coping strategies toward influence of various environmental policies, the fluctuation process of PM₂.₅ concentrations between provinces has typical category features. In order to interactively display the disparity of fluctuation process, we have taken the overall mean function of China as the benchmark for comparison (blue dashed line), Figure 13 displays how the PM₂.₅ concentration functions vary from province to province, with the mean function of each category in a red solid line. From the perspective of absolute level, we can see the average value of PM₂.₅ concentration for the sixth, the seventh and the eighth category far outweighed the overall mean and their highest value occurred around 2007. However, the average value of the second, the fourth and the fifth group is far less than the overall mean, especially the fifth group which exhibited nearly a horizontal fluctuation trajectory, meaning that there were almost no substantial changes in the PM₂.₅ concentration fluctuations. The mean curves of the first and the third category seemed to be overlapping with the trajectory of the overall mean, indicating that the level of PM₂.₅ concentration for the two categories represented the overall situation of PM₂.₅ concentration in China.

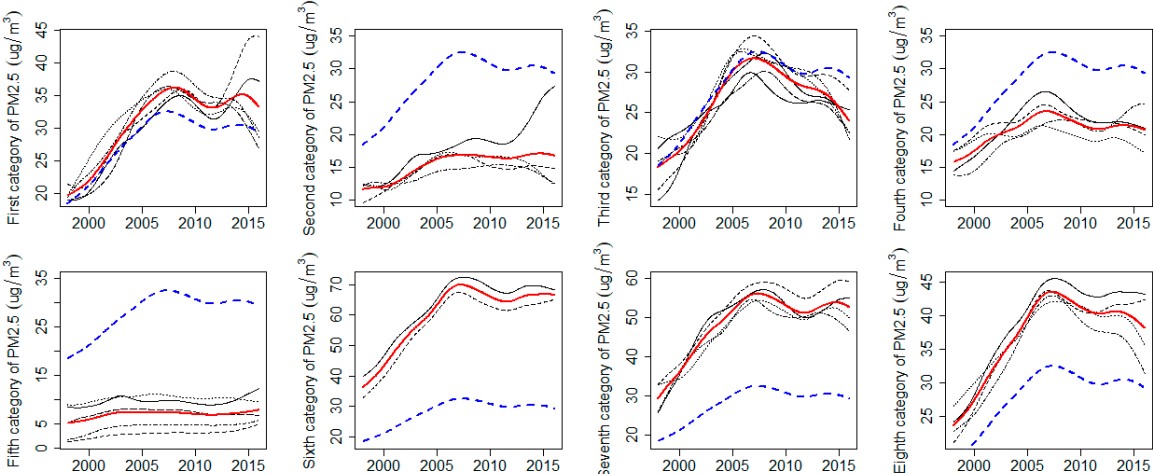

**Figure 13.** Mean curves of eight groups (**red**) with the benchmark of national average (**blue**).

Since the PM₂.₅ concentrations usually originate from multiple sources, besides motor vehicle usage and static atmosphere flow, we focus on tracking the major cause for regional difference in PM₂.₅ concentration from the perspectives of industrial activities and energy structures. According to the spatial distribution of each group in Figure 9 and data from the "Statistical Yearbook of China (1998–2016)" [58], we found that the provinces with highest level in groups six, seven and eight were mainly located in the Beijing-Tianjin-Hebei region and the Yangtze River Delta region, as well as their surrounding provinces. As is well known, the above regions are the leaders in social and economic development in China, and their prosperity was established on the massive consumption of fossil fuels (coal and oil), especially in colder seasons. The sources of PM₂.₅ in Yangtze River Delta could be attributed to the secondary pollution and active economic activities. Actually, most of traditional manufacturing industries, such as electronics industry and transportation service, located at Yangtze River Delta in China, and a large labor force including ordinary workers and high-tech talent resides in this region. The labor-intensive industries whose layout focused on upstream and intermediate products of industrial chains, produced large quantities of volatile organics, which are the main components of PM₂.₅ in Yangtze River Delta. Besides, the global night-time light data from 1992 to 2012 indicates that the Yangtze River Delta is still the most active economic area [59]. According to environmental statistics from 1998 to 2016 [60], the proportion of fumes, such as SO2 and NOx, discharged from vehicles is closing in on that from factories, and have an exceeding tendency. After chemical reactions in atmosphere, the fumes transmuted into smaller particulate pollutants, such

as sulphates and nitrate. Although the pollutants from factories are declining due to the campaign of "Desulphurization and Denitrification" launched in all industrial sectors, the growing number of vehicles is increasing the emission of pollutants in the Yangtze River Delta of China.

In contrast, the provinces with lowest $PM_{2.5}$ concentrations in the second, fourth and fifth group mainly located in two kinds of regions, that are the provinces of tourism and regions in western China. We can see that the fifth group was mainly composed of frontier provinces in western China, which is a major exporter of labor force due to its low economic development or its short industrial chain. It should be noted that the trajectories of $PM_{2.5}$ concentration in the fifth group is almost horizontal with constant deviations. The reason for this is that their highly homogenous economic development was supported by traditional agriculture and livestock farming. Thus, the level of $PM_{2.5}$ concentration in the fifth group is the lowest, seldom effected by adjustments of the industrial structure. Different to provinces in the fifth group, tourism is the pillar industry of provinces in the second group. In order to keep appealing to tourists with their beautiful environments, these provinces have to adopt environmentally-friendly sustainable economic development modes. However, the improving economical development of the second group as well as their comfortable living environment, attracts more and more residents and results in a growing quantity of vehicles. Thus, the $PM_{2.5}$ concentrations of the second group exhibit a slowly growing trend, with an increasing deviation. As for provinces in the fourth group, the $PM_{2.5}$ concentration of Fujian and Taiwan are closely related to human activity and highly developed manufacturing industries. Located at the southeast coast of China and strongly influenced by maritime monsoon, it is hard to form high concentrations of particle pollution in Fujian and Taiwan. As for Shanxi and Ningxia, the main source of $PM_{2.5}$ is dust aerosols resulting from soil erosion and the smoke discharged from energy bases. Due to their open topography, the pollutants of Shangxi and Ningxia can rapidly diffuse due to being influenced by the stable atmospheric circulation in these regions. Except for differences in fluctuation amplitude, the time of turning points corresponding to the fourth group is consistent with that of the national mean, meaning that provinces in the fourth group can adjust their industrial structures quickly according to environmental protection policies.

The $PM_{2.5}$ concentrations of provinces in the first and third group represents the average level and dominant (tendency) of China. These provinces can be classified into two categories, one category located in northeast China is characterized as developed heavy industry, such as Liaoning and Jilin. The other category is located in southeast China with the highest population density, including Chongqing and Hong Kong. The region classification in this paper indicates that the spatial distribution of $PM_{2.5}$ concentration has obvious characteristics of spatial agglomeration. Besides, the classification of $PM_{2.5}$ concentration for 34 provinces in our study is basically consistent with the regional definition, "three districts and ten groups", of 12th Five-Year Plan for Air Pollution Prevention and Control in Key Regions jointly issued by the Ministry of Ecology and Environment, the State Development and Reform Commission and the Ministry of Finance of China [61].

In order to further analyze the differences in the growth of $PM_{2.5}$ concentrations from dynamic perspectives, which also is the advantage of FDA, we plot the trajectories of velocity and acceleration for eight groups in Figure 14. Upon the comparison of fluctuation trajectories between every group, it can be found that the provinces in the sixth group not only possess the largest level of $PM_{2.5}$ concentration, but their fluctuation amplitudes of velocity and acceleration are also the largest ones. Besides, by comparing the turning points in the curves of velocity and acceleration with the issued time of environment protection policies, we found that the provinces in the sixth group could adjust their industrial structure and pollution emissions in time in accordance with the policy requirements. The absolute level and amplitudes of velocity and acceleration for $PM_{2.5}$ concentration of the seventh and eighth group ranked the second and the third, respectively, and their turning points are also highly concurrent with the issued time of environment protection policies. Compared to the regions with the highest $PM_{2.5}$ concentration, the amplitudes of velocity and acceleration for $PM_{2.5}$ concentration of the

second, the fourth and the fifth group were remarkably small, but there was few turning points at the issued time of environment protection policies.

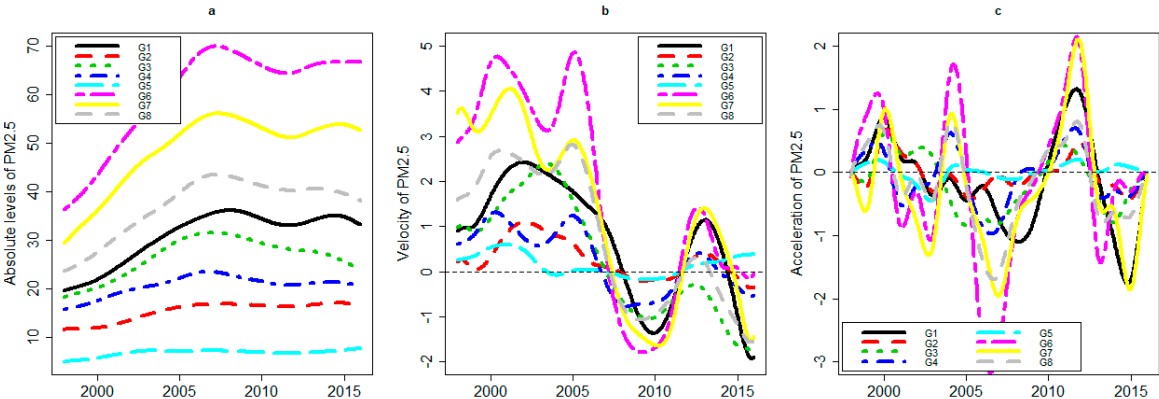

**Figure 14.** (**a**) PM$_{2.5}$ concentration functions with (**b**) firs-order and (**c**) second-order derivatives of eight groups for the years from 1998 to 2016.

The dynamic analysis of PM$_{2.5}$ concentration indicates that, although the environment protection policies issued by government sectors in China could have dramatic influence on reducing the overall PM$_{2.5}$ concentration, especially in the high pollution regions, the rebound effect would also be obvious after the control periods of regulations. However, the regulating effect of policies was negligible in the low pollution regions because of their environmentally friendly economic development modes. The implication of our empirical results is that the relationship between China's existing economy development mode and environmental protection is still in an irreconcilable stage, and it is hard to eliminate or reduce PM$_{2.5}$ concentrations by just relying on the government's administrative intervention. As low pollution areas have the subjective motivation of protecting the environment to sustain their pillar industry, the government should fundamentally devote its efforts to reducing pollution levels in high pollution areas.

## 4. Conclusions and Discussion

As a developing country with vast territory and a typical dual economic structure, the rapid development of China occurs at the expense of environment and energy, which has resulted in serious air pollution. Accurately identifying the spatial and temporal patterns of haze pollution is a prerequisite for rational formulation and effective implementation of haze control policies. This study employed FDA techniques to represent PM$_{2.5}$ concentration data in the form of a smoothing curve for each province. Based on the continuous curves reconstructed from discrete noisy PM$_{2.5}$ concentration data with roughness penalty, the FPCA was adopted to decompose the temporal variability of PM$_{2.5}$ concentration curves, and the patterns of PM$_{2.5}$ concentration in 34 provinces was determined using adaptive weighting clustering analysis. The analysis continued with a functional ANOVA to verify the significance of differences between eight groups, and with further exploration in their spatial differences, both from static and multiple dynamic perspectives. The conclusions with policy implications obtained from this study are as follows.

(1) Imposing roughness penalty on the curves' reconstruction of PM$_{2.5}$ concentration could emphasize the dominant trend of fluctuation, thus enhancing the interpretability of variability implied in PM$_{2.5}$ concentration curves. The standard deviation trajectory of PM$_{2.5}$ concentrations perfectly followed the growing pattern of the overall mean function, which means that facing the opportunity for rapidly developing economy at the expense of environment pollution, the decision-making of different provinces differed vastly, whether for subjective reasons of excessively pursuing GDP or for objective reasons of industrial structure and resource endowment. The above conclusions imply that quite a few provinces could rationally balance extensive

economic development with ecological sustainability. Consequently, the feasible approach to eliminate haze pollution should emphasis on optimizing, upgrading and transferring of industrial structure. In particular, the government should encourage low pollution regions, through cutting their taxes or increasing their subsidies, to sustain their environmentally-friendly economic development.

(2)　The temporal variability of $PM_{2.5}$ concentration from 1998 to 2016 could be decomposed into three distinctive sub-fluctuation modes by FPCA, which depicts the variations in the beginning, the middle interval and the end of the research period, respectively. Remarkably, the middle interval with largest variation portrayed by the first FPC perfectly matches with the period of the "ten golden years" for coal, and the variance contribution rate of the first FPC far outweighs that of the other two, meaning that the fluctuation of $PM_{2.5}$ concentrations for 34 provinces was mainly located at the period of extensive economic growth. The empirical result again verifies the different coping strategies among the 34 provinces when facing the choice of developing the economy at expense of the environment and energy. The contribution to empirical methodology derived from this study is that the huge disparity in classification information among the three FPCs requires different weights when conducting clustering analysis on 34 $PM_{2.5}$ concentrations curves. Therefore, the same inputs or approaches might not be useful in modeling the pollution processes for different regions.

(3)　The fluctuation patterns of $PM_{2.5}$ concentration functions were classified into eight groups via adaptive weighting cluster analysis, and the effect of spatial and geographical locations was analyzed using functional ANOVA. The test results indicate that the differences between the eight groups was significant, whether from the static perspective or dynamic potential. The reason of differences in the $PM_{2.5}$ concentration patterns could possibly be due to the effect of geographic and industrial factors, as well as the different coping strategies of environmental policies. Multiple comparisons of fluctuation patterns show that the heavy pollution areas not only have the highest level of $PM_{2.5}$ concentration, but also have the largest longitudinal amplitude of velocity and acceleration. The tuning points of $PM_{2.5}$ concentration curves for the heavy pollution areas highly matched the issued time of environmental policies, whereas the effect of environmental policies in low pollution areas was not obvious. The findings reveal that the characteristics of $PM_{2.5}$ concentration are very dependent on the industrial structures of the provinces. As such, it is hard to eliminate haze pollution by relying solely on the government's administrative intervention. Thus, the direct way of reducing $PM_{2.5}$ concentration in the short term is to maintain the continuity of environmental policies. In the long run, how to encourage enterprises to transform or upgrade industrial structure via revenue decrease or financial subsidy is an important and unavoidable issue for government to eliminate haze pollution fundamentally.

Compared with the existing literature, the main contribution of this study is focused on how the FDA technique can be used for $PM_{2.5}$ concentrations data analysis. This paper has significance for both empirical methodology and important policy implications. Instead of utilizing discrete noisy $PM_{2.5}$ concentration data, we can create a functional form for the data which could be analyzed over any time interval. So we are able to extract additional information contained in the smoothing curve and its derivatives which may not be normally available from traditional statistical methods. The findings from this study, such as significant differences in $PM_{2.5}$ concentration patterns between regions, not only provide a guideline for analyzing the effectiveness of current air quality control regulations, but also provide information for the environment management for provinces, as well as suggestions on sustainable development for China's government. As a future research direction, significant differences in $PM_{2.5}$ concentration patterns between regions signify that a different approach in modeling the process should be employed, especially linking the change of $PM_{2.5}$ concentration to policy-related implications using functional concurrent models.

**Author Contributions:** Conceptualization, D.W. and L.H.; Methodology, D.W.; Software, D.W. and K.B.; Validation, D.W., Z.Z. and L.H.; Formal analysis, Z.Z.; Investigation, K.B.; Resources, D.W. and K.B.; Data curation, R.L.; Writing—original draft preparation, D.W.; Writing—review and editing, L.H.; Visualization, D.W.; Supervision, L.H.; Project administration, D.W.; Funding acquisition, D.W. and L.H.

**Funding:** This study was supported by the Fundamental Research Funds for the Central Universities (Project nos. 2015WA01).

**Acknowledgments:** We thank the editor and anonymous referees for helpful comments. All errors are our own.

**Conflicts of Interest:** No potential conflict of interest was reported by the authors.

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
