# Peer review of "Spatial and Temporal Variabilities of PM2.5 Concentrations in China Using Functional Data Analysis"

_sustainability, doi:10.3390/su11061620_

Round 1
Reviewer 1 Report
While I appreciate the empirical approach of the authors with respect to determining the temporal variability of PM2.5 concentrations ,of 34 Chinese provinces, using FDA, I am afraid that the implications of this paper may be limited in application especially in this Journal. In particular, I raise the following comments:
1) The data utilized in the study are those from the MODIS AOD,and not the actual monitoring station data. In such case, the authors will be faced with two noise: a) noise coming from the AOD-PM2.5 (affected by cloud cover, etc.), and b) random noise from those existing even in the background station (affected by wind speed, etc.). I am not particularly confident that FDA controlled for both types of noises.
2) Income was used to determine the optimal GCV ctiterion. How about other factors? If there were other factors (population density, green space, carbon inventory, etc.), GCV may change.
3) The use of FDA to empirically show the change of PM2.5 concentration with time may only be limited to the velocity and trajectory of these changes, temporally. I think it would be insufficient, in context, to link it to policy-related implications. PM2.5 is a composite material with many components from various emission sources. In this study, there were no specific policies examined, so to say that the speed of these changes in PM2.5 without providing thorough results is moot.
Author Response
We have carefully revised the manuscript in accordance with the reviewer’s comments, and proof-read the manuscript to minimize typographical, grammatical, and bibliographical errors. We acknowledge the reviewer’s comments and suggestions very much, which are valuable in improving the quality of our manuscript. Now we answer the questions one by one. The responses to the reviewer’s comments are the following. Point 1: The data utilized in the study are those from the MODIS AOD, and not the actual monitoring station data. In such case, the authors will be faced with two noise: a) noise coming from the AOD-PM2.5 (affected by cloud cover, etc.), and b) random noise from those existing even in the background station (affected by wind speed, etc.). I am not particularly confident that FDA controlled for both types of noises. Response 1: We greatly appreciate the reviewer’s comment and constructive feedback. In accordance with your suggestions, we added the explanation of data sources. The data sets used here are obtained from the study by van Donkelaar et al.(2016) [1], which had been calibrated each AOD source using AERONET observations. More information regarding your doubts in terms of these noise can be found in reference[2]. Actually, the research based on satellite-based monitoring data has won the recognition of the academics, which owed to the works of Nordhaus et al.[3,4] who won the Nobel Prize Economics in 2018. Thus, the satellite-based monitoring data employed by this study is reliable. Additionally, from the technical perspective of empirical analysis, functional data analysis (FDA) owns the congenital advantage of modeling noisy data when smoothing with roughness penalty [5,6], even the data is sparse or sampled unequally. Thus, combined the reliable data source with the advanced methodology, it is reasonable to draw reliable conclusions. The added explanation to the data was marked in red in the text. References 1: [1] Donkelaar A V, Martin R V, Brauer M, et al. Global estimates of fine particulate matter using a combined geophysical-statistical method with information from satellites, models, and monitors[J]. Environmental science & technology, 2016, 50(7): 3762-3772 [2] Donkelaar A V , Martin R V , Brauer M , et al. Use of Satellite Observations for Long-Term Exposure Assessment of Global Concentrations of Fine Particulate Matter. Environmental Health Perspectives, 2014, 123(2): 135-143. [3] Nordhaus W D . Geography and Macroeconomics: New Data and New Finding. Proceedings of the National Academy of Sciences, 2006, 103(10):3510-3517. [4] Chen X; Nordhaus W D. Using luminosity data as a proxy for economic statistics. Proceedings of the National Academy of Sciences, 2011, 108(21): 8589-8594. [5] Tsay R S . Some Methods for Analyzing Big Dependent Data. Journal of Business & Economic Statistics, 2016, 34(4):673-688. [6] Craven P , Wahba G . Smoothing noisy data with spline functions. Numerische Mathematik, 1978, 31(4):377-403. Point 2:Income was used to determine the optimal GCV criterion. How about other factors? If there were other factors (population density, green space, carbon inventory, etc.), GCV may change. Response2: We are terribly sorry for misusing an variable when plotting Figure 1. The correct label for Y-axis in Figure 1 should be "Generalized cross validation (K)" rather than " Generalized cross validation (Income)". In other words, the GCV (k) is a function with respect to the smoothing parameter k, that is , as presented in formula (2). We greatly appreciate the reviewer’s comments on the optimal GCV criterion. In fact, how to select the smoothing parameter via generalized cross-validation (GCV) is a mature technique in nonparametric statistics, which is proposed by Craven P and Wahba G .(1978) [1]. Ramsey et al. (2005) introduced the GCV criterion to the framework of functional data analysis (FDA) [2,3]. The role of the optimal smoothing parameter, determined by minimizing the GCV, is to give a reasonable tradeoff between the goodness of fit and the dominant trend when reconstructing intrinsic functions from discrete noisy data. In FDA, quantitative variables (such as population density) sampled at different time points will be converted to continuous functions with fluctuation, while qualitative factors (such as regions) kept invariable will be converted to constant functions without any roughness. It should be noted that every intrinsic function is reconstructed separately, thus, the GCV corresponding to different factors may be varied. The complete theoretical review of GCV can be referred in [1,4]. We updated the correct variable of Figure 1 in this revised manuscript. References 2: [1] Craven P , Wahba G . Smoothing noisy data with spline functions. Numerische Mathematik, 1978, 31(4):377-403. [2] Ramsay, J.O.; Silverman, B.W. Functional Data Analysis, 2nd ed.; Springer: New York, NY, USA, 2005. [3] Ramsay, J.O.; Hooker, G.; Graves, S. Functional Data Analysis with R and MATLAB; Springer: New York, NY, USA, 2009. [4] Kokoszka, P.; Reimherr, M. Introduction to Functional Data Analysis; Chapman and Hall/CRC Press: London, UK, 2017. Point 3:The use of FDA to empirically show the change of PM2.5 concentration with time may only be limited to the velocity and trajectory of these changes, temporally. I think it would be insufficient, in context, to link it to policy-related implications. PM2.5 is a composite material with many components from various emission sources. In this study, there were no specific policies examined, so to say that the speed of these changes in PM2.5 without providing thorough results is moot. Response3: We sincerely thank the reviewer's suggestion, and we fully acknowledge the importance of finding out more reasons for the change of PM2.5 concentration in China. Just as you said, PM2.5 is a composite material with many components from various emission sources, such as sea spray, road dust, soil, motor vehicle usage, industrial activities, domestic activities, and biomass burning. Thus, understanding the behaviour of PM2.5 concentration is becoming more important in air pollution investigation. To the best of our knowledge, numerous academic papers dedicated to find effective measures to eliminate or reduce its emission level, whereas there was no consensus. One of the main reasons could be traceable to the short of critical understanding the spatial and temporal variability of PM2.5 concentrations, especially from the dynamic perspectives. It is known to all, under the typical dual economic structure, the PM2.5 concentrations in China has obvious regional differences[1,2], and the impact of environmental protection policies exhibited significant time lags [3-5]. Besides, most of the existing studies focused on depicting or modelling the fluctuations of PM2.5 concentration using traditional statistical methods upon discrete noisy data, which requires high data quality such as sampling at equal frequencies. However, seldom studies had taken into account the dynamic characteristics of PM2.5 concentration. As a result, the homogenous environmental protection policies issued by government lacked pertinence for different regions, and were not good enough in time-sensitive. We fully agree with the reviewer's suggestion that we should examine the relationship between specific policies and the speed of changes in PM2.5. In fact, we have begun to test the significance of relationship between the two using functional concurrent models in our next article. However, every quantitative methods aimed at analyzing the relationship between the two are dependent on the deep recognition of their spatial-temporal characteristics. The main concern of this study is to build a functional data object from discrete PM2.5 observations by looking at how PM2.5 concentration fluctuates, both spatially and temporally, in the form of smoothing curves. The FDA methods employed in this study are able to extract additional information contained in the function and its derivatives which may not be normally available from traditional statistical methods. By converting discrete PM2.5 concentration values into a smoothing curve with roughness penalty, the continuous process of PM2.5 concentration for each province was presented. Functional concepts such as functional descriptive statistics and functional analysis of variance were applied to describe the spatial and temporal PM2.5 concentration variations in different provinces and at any time throughout the year. The variance decomposition via functional principal component analysis indicates that the highest mean and largest variability of PM2.5 concentration located at the period from 2003 to 2012, during which national environmental protection policies issued intensively. Whereas the beginning and end stages own the equal information of variability, which was far less than that of the middle stage. Since the PM2.5 concentration curves showed different fluctuation patterns in each province, the adaptive clustering analysis combined with functional analysis of variance were adopted to explore the categories of PM2.5 concentration curves. The classification result shows that: (1) There existed eight patterns of PM2.5 concentration among 34 provinces, and the difference among different patterns was significant whether from static perspective or multiple dynamic perspectives. (2) Air pollution in China presents an characteristic of high-emission "club" agglomeration. Comparative analysis of PM2.5 profiles showed that the heavy pollution areas could timely adjust their emission level according to the environmental protection policies. Whereas low pollution areas characterized by tourism industry would rationally treat the opportunity of developing economy at the expense of environment and resources. As a developing country with vast territory and typical dual economic structure, the rapid development of China is at expense of environment and energy, which resulted in serious air pollution. As a preliminary exploration of air pollution research in "Air Quality Assessment Standards and Sustainable Development in Developing Countries" which is a special issue of Sustainability, we are interested in looking at how the spatial and temporal patterns of the yearly variation of PM2.5 in China can be explained by its geographical regions or by any other factor such as industrial structure, and if possible, this analysis could make a significant contribution to policymakers in future impact studies. Compared with the existing literatures, This paper is of significance for both empirical methodology and important policy implications. The findings from this study, such as significant differences in PM2.5 concentration patterns between regions, not only provide a guideline for analyzing the effectiveness of current air quality control regulations, but also provide information to environment management for provinces, as well as suggestions to sustainable development for China's government. It is no exaggeration to say that significance test of the linkage between the tuning points of PM2.5 concentration curves with the issuing time of environmental protection policies is of great importance. Whereas, the detection of tuning points of PM2.5 concentration curve is via comparing the positive-negative alternation of its velocity and trajectory. We fully acknowledge the importance of linking the change of PM2.5 concentration to policy-related implications, which is the key issues in our next paper we have basically completed. However, as a preliminary work, the main purpose of this paper is introducing an advanced technique, FDA, to study the spatial and temporal characteristics of PM2.5 concentration in China based on the continuous and dynamic perspectives. Thus, we just give a rough description and explanation of policy-related implications in this study. Last but not the least, we sincerely thank the reviewer's constructive comments which is the core content of our future work, and we added the valuable suggestions in the discussion part. References 3: [1] Guan D, Su X, Zhang Q, et al. The socioeconomic drivers of China’s primary PM2.5 emissions. Environmental Research Letters, 2014, 9(2):024010. [2] Wang Y Q, Zhang X Y, Sun J Y, et al. Spatial and temporal variations of the concentrations of PM10, PM2.5 and PM1 in China. Atmospheric Chemistry and Physics, 2015, 15(23):3585–13598. [3] Cheng N , Zhang D , Li Y , et al. Spatio-temporal variations of PM2.5 concentrations and the evaluation of emission reduction measures during two red air pollution alerts in Beijing. Scientific Reports, 2017, 7(1):8220. [4] Peng, J.;Chen, S. et al. Spatiotemporal patterns of remotely sensed PM2.5 concentration in China from 1999 to 2011. Remote Sensing of Environment, 2016, 174:109–121. [5] Shao, S.; Li, X.; Cao, J. H.; Yang L. L. China's Economic Policy Choices for Governing Smog Pollution Based on Spatial Spillover Effects. Economic Research Journal, 2016, (9):73-88.

Reviewer 2 Report
The fluctuation patterns of PM2.5 concentration between provinces were compared from 1998 to 2016 in China, both spatially and temporally, within the framework of functional data analysis. The discrete time point values are considered as observations of continuous functions over a continuum and the Functional Data Analysis (FDA) is applied. The PM2.5 concentration at each province was presented by converting the discrete PM2.5 concentration values into a smoothing curve with roughness penalty. The analysis indicates that the highest mean and largest variability of PM2.5 concentration was from 2003 to 2012. In the beginning and end stages the variability was similar but in the middle stage it was not the case. Since the PM2.5 concentration curves showed different fluctuation patterns in each province, the adaptive clustering analysis combined with functional analysis of variance were adopted to explore the categories of PM2.5 concentration curves. The classification result showed that there existed eight patterns of PM2.5 concentrations among the 34 provinces, and the difference among the different patterns was significant. Comparative analysis of PM2.5 profiles showed that the heavy pollution areas agreed with the emission levels according to the environmental protection measures. The low pollution areas, which are characterized by the tourism industry, provide the opportunity of developing economy by protection of environment and resources.
General comments
It is concluded that this study provides empirical information for China to reduce or eliminate the haze pollution fundamentally. This can be supported because this type of statistical investigation of the spatial and temporal variation of PM2.5 provides a very valuable connection between economic development, PM and other air pollutant emissions and air pollution exposure.
The paper addresses relevant scientific questions. The paper presents novel concepts, ideas and tools.
The scientific methods and assumptions are valid and clearly outlined so that substantial conclusions are reached.
The description of experiments and calculations are sufficiently complete and precise to allow their reproduction by fellow scientists.
The quality, information and caption of the figures are good.
The related work is well cited.
Title and abstract reflect the whole content of the paper. The abstract should be focused more on the results of this study.
The overall presentation is well structured and clear. The language is fluent but can be improved in some parts.
The abbreviations and units are generally correctly defined and used.
Specific Comments
The figure captions should be understandable by themselves – e.g. abbreviations should be explained.
Not all terms are defined correct – e.g. line 154 PENSSE.
The references are written very different but should follow the journal guidelines.
Technical corrections
Numbers in some figures are too small to read – e.g. Figs. 5, 6, 7, 10, 11, 12.
Author Response
First of all, we sincerely send our thanks to the reviewer for his/her approval of our work. We acknowledge the reviewer’s comments and suggestions very much, which are valuable in improving the quality of our manuscript. Now we answer the questions one by one.
Point 1: General comments: It is concluded that this study provides empirical information for China to reduce or eliminate the haze pollution fundamentally. This can be supported because this type of statistical investigation of the spatial and temporal variation of PM2.5 provides a very valuable connection between economic development, PM and other air pollutant emissions and air pollution exposure. The paper addresses relevant scientific questions. The paper presents novel concepts, ideas and tools. The scientific methods and assumptions are valid and clearly outlined so that substantial conclusions are reached. The description of experiments and calculations are sufficiently complete and precise to allow their reproduction by fellow scientists. The quality, information and caption of the figures are good. The related work is well cited. Title and abstract reflect the whole content of the paper. The abstract should be focused more on the results of this study. The overall presentation is well structured and clear. The language is fluent but can be improved in some parts. The abbreviations and units are generally correctly defined and used.
Response 1: Many thanks, your affirmation and acceptance for this study solidified our faith in continuing the research of air pollution in developing countries. According to your suggestions, we focus the abstract on the core results of this study, and proof-read the manuscript to minimize typographical, grammatical, and bibliographical errors.
Point 2: Specific Comments: The figure captions should be understandable by themselves – e.g. abbreviations should be explained.
Not all terms are defined correct – e.g. line 154 PENSSE.
The references are written very different but should follow the journal guidelines.
Response 2: We are sorry for using abbreviations in the figures. In this revised manuscript, we have improved the understandability for every Figure caption, such as replacing GCV with generalized cross-validation. As to the term of PENSSE, its full name is "penalized residual sum of squares" or " sum of squared fitting residuals for the roughness penalty" which is named by Ramsey et al.(2006) in their famous works [1,2], and many studies directly took this abbreviation as a well-known term in FDA, such as [3]. For the preciseness and accuracy of this manuscript, we give the full name of every term for its first citation, and present their abbreviations in the parentheses after their first citations. We array the references by the order of their advent, and we will give a thorough correction follow the journal guidelines in the last revision.
References 1:
[1] Ramsay, J.O.; Silverman, B.W. Functional Data Analysis, 2nd ed.; Springer: New York, NY, USA, 2005.
[2] Ramsay, J.O.; Hooker, G.; Graves, S. Functional Data Analysis with R and MATLAB; Springer: New York, NY, USA, 2009.
[3] Suhaila, J.; Jemain, A. A.; Hamdan, M. F. Comparing rainfall patterns between regions in Peninsular Malaysia via a functional data analysis technique. J. Hydrol. 2011, 411, 197–206.
Point 3:Technical corrections
Numbers in some figures are too small to read – e.g. Figs. 5, 6, 7, 10, 11, 12.
Response 3: Sorry for our carelessness. We have enlarged the numbers in relevant figures, if impossible, we will provide all the original graphs with high quality to the assistant editor.

Reviewer 3 Report
Dear Authors,
The paper "Spatial and Temporal Variabilities of PM2.5 Concentrations in China using Functional Data Analysis" presents a novel statistical methodology to evaluate PM2.5 concentrations in serveral obervation points. The topic of the paper is appropiate for the journal and it could be accepted after the following considerations:
1) The introduction is well written but i suggest to consider more other methodological approaches that usually work weel with large scale pollution dispersion. I suggest to consider Lnd use regressions models and refer to them. Please consider the following papers as suggestion for the bibliography:
- Famoso F, et al. 2017. Measurement and modeling of ground-level ozone concentration in Catania, Italy using biophysical remote sensing and GIS. International Journal of Applied Engineering Research, 12 (21), 10551-10562.
- Wu CD et al. 2018. A hybrid kriging/land-use regression model to assess PM2.5 spatial-temporal variability. Science of the Total Environment, 645, 1456-1464.
- Mukherjee, et al. 2018. Assessment of local and distant sources of urban PM2.5 in middle Indo-Gangetic plain of India using statistical modeling. Atmospheric Research, 213, 275-287.
- Brokamp, C et al. 2017. Exposure assessment models for elemental components of particulate matter in an urban environment: A comparison of regression and random forest approaches. Atmospheric Environment, 151, 1-11.
2) Please use a nomenclature for all variables.
3) Please put units of measurement in each graph.
4) An Extensive editing of English language and style are required. There are several grammatical error. Please consider the idea to contact a professional english translator.
After this revisions the paper will be accepted.
Kind regards
Author Response
We have carefully revised the manuscript in accordance with the reviewers’ comments, and proof-read the manuscript to minimize typographical, grammatical, and bibliographical errors. We acknowledge the reviewer’s comments and suggestions very much, which are valuable in improving the quality of our manuscript. Now we answer the questions one by one. The responses to the reviewers’ comments are the following.
Point 0: The paper "Spatial and Temporal Variabilities of PM2.5 Concentrations in China using Functional Data Analysis" presents a novel statistical methodology to evaluate PM2.5 concentrations in several observation points. The topic of the paper is appropriate for the journal and it could be accepted after the following considerations:
Response 0: We sincerely send our thanks to the reviewer for his/her approval of our work.
Point 1: The introduction is well written but I suggest to consider more other methodological approaches that usually work well with large scale pollution dispersion. I suggest to consider Land use regressions (LUR) models and refer to them. Please consider the following papers as suggestion for the bibliography:
- Famoso F, et al. 2017. Measurement and modeling of ground-level ozone concentration in Catania, Italy using biophysical remote sensing and GIS. International Journal of Applied Engineering Research, 12 (21), 10551-10562.
- Wu CD et al. 2018. A hybrid kriging/land-use regression model to assess PM2.5 spatial-temporal variability. Science of the Total Environment, 645, 1456-1464.
- Mukherjee, et al. 2018. Assessment of local and distant sources of urban PM2.5 in middle Indo-Gangetic plain of India using statistical modeling. Atmospheric Research, 213, 275-287.
- Brokamp, C et al. 2017. Exposure assessment models for elemental components of particulate matter in an urban environment: A comparison of regression and random forest approaches. Atmospheric Environment, 151, 1-11.
Response 1: We greatly appreciate the reviewer's important and constructive comments. Your feedback made us clear the necessity of providing sufficient background in the introduction, especially methodological approaches that work well with large scale pollution dispersion. According to your valuable suggestions, we download the relevant references and read them carefully. Just as the reviewer's assertion, the methodological approaches in the relevant references are advanced and bear universal applicability. Taken the above references as comparison benchmarks, we first summarize the framework of models of relevant references, then compare our method with the existed works, and add the merits of our methodology in the introduction. The important papers recommended by the reviewer are added in the list of references. The added analysis were marked in red in the text.
Point 2&3: Please use a nomenclature for all variables. Please put units of measurement in each graph.
Response2&3: Many thanks for your constructive feedback. We improve the nomenclature for all variables and put units of measurement in each graph.
Point 4: An Extensive editing of English language and style are required. There are several grammatical error. Please consider the idea to contact a professional English translator. After this revisions the paper will be accepted.
Response 4: We have read the manuscript carefully to minimize typographical, grammatical, and bibliographical errors.

Reviewer 4 Report
The manuscript of Wang et al. is suitable for publication in “Sustainability”. The study presented and developed is interesting for the scientific community, the methods were well chosen and applied and the text is self-explanatory.
There are some notes to take into consideration:
- Line 14: “one of the serous” serous should be corrected; and in line 15 pollutant should be plural.
- Line 27-39: after (1) it should be “there” instead of “There” and before (2) it should be used a “;”. Also, after (2) it should be used lowercase letter.
- Sometimes 2.5 is subscripted (line 133), while others it is not (125, 384, 397, 399, …)
- Why reference [40] is in red in line 210? There is not a very good use of the reference style, especially in section 2.1.
- Line 346 “In light of the that , the vast…” the space after “that” should be removed
- Table 1 – pay attention to the commas “,”
- Line 407 instead of figure should be Figure
- Line 411: the first part of this sentence needs a verb “Whereas the average value of the second, the fourth and the fifth group far less than the overall mean, …”
- What is the need for Shanxi to be written in caps lock or to be written as Shangxi (line 460)?
- Line 490: instead of “could imposed”, should be “could impose”
- Line 557: literature should be singular
- Section 3.5 and Conclusions can be shortened.
The authors should choose between UK or USA English.
Author Response
We have carefully revised the manuscript in accordance with the reviewers’ comments. We acknowledge the reviewer’s comments and suggestions very much, which are valuable in improving the quality of our manuscript. Now we answer the questions one by one. The responses to the reviewers’ comments are the following.
Point 0: The manuscript of Wang et al. is suitable for publication in “Sustainability”. The study presented and developed is interesting for the scientific community, the methods were well chosen and applied and the text is self-explanatory.
Response 0: We sincerely send our thanks to the reviewer for his/her approval of our work.
Point 1: Line 14: “one of the serous” serous should be corrected; and in line 15 pollutant should be plural.
Response 1: We have made corrections according to the reviewer's advice and requirement.
Point 2: Line 27-39: after (1) it should be “there” instead of “There” and before (2) it should be used a “;”. Also, after (2) it should be used lowercase letter.
Response2: Many thanks for your detailed scrutiny, we correct all the formatting errors.
Point 3: Sometimes 2.5 is subscripted (line 133), while others it is not (125, 384, 397, 399, …)
Response 4: Many thanks for your detailed scrutiny, we correct all the formatting errors.
Point 5: Why reference [40] is in red in line 210 ? There is not a very good use of the reference style, especially in section 2.1.
Response 5: Sorry for our carelessness, all the words should be black. We have made corrections following the journal guidelines, and will continuing correct if necessary.
Point 6: Line 346 “In light of the that , the vast…” the space after “that” should be removed
Response6: Sorry for our carelessness, we delete the space after “that”.
Point 7: Table 1 – pay attention to the commas “,”
Response 7: We consolidate the format of commas “,” in Table 1, and add the full stop.
Point 8: Line 407 instead of figure should be Figure
Response 8: Sorry for our carelessness, we replace figure by Figure.
Point 9: Line 411: the first part of this sentence needs a verb “Whereas the average value of the second, the fourth and the fifth group far less than the overall mean, …”
Response9: Sorry for our carelessness, we add "is" before " far..".
Point 10: What is the need for Shanxi to be written in caps lock or to be written as Shangxi (line 460)?
Response 10: There are two different provinces with the same pronunciation "shanxi". In order to distinguish them, we write one of them in caps lock.
Point 11&12: Line 490: instead of “could imposed”, should be “could impose”. Line 557: literature should be singular.
Response11&12: Many thanks. We made corrections according to your suggestions.
Point 13: Section 3.5 and Conclusions can be shortened.
Response13: We shortened the conclusions according to the reviewer's suggestions.

Round 2
Reviewer 1 Report
The authors have addressed my queries. However, I would have particularly appreciate it if you could have included the explanations within the manuscript, as sort of additional explanation. The extensive response to the queries merit inclusion of the parts of such response to the manuscript.
Author Response
We have carefully revised the manuscript in accordance with the reviewers’ comments. The responses to the reviewers’ comments are the following.
Response to Reviewer 1 Comments
Point 1: The authors have addressed my queries. However, I would have particularly appreciate it if you could have included the explanations within the manuscript, as sort of additional explanation. The extensive response to the queries merit inclusion of the parts of such response to the manuscript.
Response 1: We greatly appreciate the reviewer’s comment and constructive feedback. In accordance with your suggestions, we added the extensive explanations within the manuscript. The added explanations was marked in red in the text. We wish our revision could meet the requirement of your advice. Last but not the least, we sincerely send our unlimited gratitude to your comments and suggestions, which are valuable not only in improving the quality of this manuscript, but also for our future research.
Response to Reviewer 2 Comments
Point 1: The changes in the manuscript are small but together with the author response the paper can be accepted.
Response 1: First of all, we sincerely send our thanks to the reviewer for your consistent help in improving the quality of our manuscript, your affirmation and acceptance for this study solidified our faith in continuing the research of air pollution in developing countries. A follow-up study will try to explore the relationship between specific policies and the changes of PM2.5. We are eager to get your valuable advice and help once more.

Reviewer 2 Report
The changes in the manuscript are small but together with the author response the paper can be accepted.
Author Response

(The authors gave the same response as above.)
